# Conventionalization of Organic Farms in Germany: An Empirical Investigation Based on a Composite Indicator Approach

**Claudia Seidel** [1,*]**, Thomas Heckelei** [1] **and Sebastian Lakner** [2] 

1   Institute for Food and Resource Economics, University of Bonn, Nussallee 21, D-53115 Bonn, Germany;
    thomas.heckelei@ilr.uni-bonn.de
2   Department of Agricultural Economics and Rural Development, University of Göttingen,
    Platz der Göttinger Sieben 5, D-37073 Göttingen, Germany; slakner@gwdg.de
*   Correspondence: claudia.seidel@ilr.uni-bonn.de

**Abstract:** The term "conventionalization" of organic agriculture was created to depict the controversially discussed phenomenon that organic agriculture departs from the core organic principles on which it is based.   We present an empirical, index-based approach to investigate developments of organic farming practices towards conventionalization.   An index of conventionalization can be used as a monitoring tool to support policymakers to further develop agricultural regulations. We calculate composite indicators for three farm types: farms specialized on crop production, farms specialized on animal husbandry, and mixed farms. Principal component analysis serves to derive objective weights based on the correlations between indicators which then allow a linear aggregation to the composite indicator.  Results show that developments towards conventionalization of the whole organic farming sector cannot be detected for German farms between 2000 and 2009.  Therefore, we do not see the necessity for changes in regulation of the organic sector with regard to conventionalization.

**Keywords:** organic agriculture; conventionalization; index; composite indicator; principal component analysis

## 1. Introduction

During the past 30 years, organic farming grew considerably. In 2018, 8.9% of the farmland in Germany was under organic production and 11.7% of the total number of German farms were organic agricultural holdings [1].  The rapid growth of the organic sector led to various transformations in organic farming. Scientific articles, as well as reports in popular media, raised concern about occurring changes within the organic sector [2–4].  Organic farming would evolve into a modified version of conventional agriculture with similar social, economic, and technical characteristics [5,6]. To denote this development, the term "conventionalization" of organic agriculture was created by Buck, Getz, and Guthman [7].  The typical examples of "conventionalized" farms are very specialized farms that buy or sell fertilizer and manure on a large scale.  Arable farms that do not keep animals and animal husbandry farms without crop production often do not maintain a closed nutrient cycle, which is an important element of organic farming [8].  Conventionalization is considered as reducing the environmental benefits of organic farming [9].

The term conventionalization is not clearly defined resulting in a partly confusing debate about this issue.  Often, developments following technical progress and stronger economic rationality in organic farming are confused with developments leading to a reduction of ecosystem services.

The debate about conventionalization of organic farming concentrates on two fundamental questions: whether the conclusions that Buck et al. [7] draw from their observations in California hold for other regions and whether conventionalization affects all farms within a region equally [10,11]. Various case studies examined the phenomenon of conventionalization in other regions. Most of those found indications of conventionalization [3,10,12–15]. However, the results were not strong and quite as diverse as the employed methodologies [11,16]. Developments often presented are as follows: smaller farms become bigger; the diversity of crops is reduced; the load of debt increases with growing capital intensification; input substitution, i.e., labor is replaced by mechanization and other industrial inputs; marketing becomes export-oriented rather than local [5,17]. Some researchers pointed to regional differences in farming, as well as political and economic environment, and argued that general results cannot be derived [6,10–12]. Case studies of the German organic farming sector did not find a common line of reasoning either. While Oppermann [1] and Best [6] uncovered tendencies towards conventionalization, Lakner and Wilken [18] stated that their results rather indicated a trend towards professionalization.

The second core issue in the debate is whether conventionalization of organic agriculture affects all farmers to the same extent or whether there might be a development towards bifurcation between large, recently converted, export-oriented farmers on the one hand, and smaller, movement-based, locally oriented farmers on the other hand [3,11,12]. The first scenario was presented by Guthman [17] who argued that involvement of agribusiness alters the conditions for all organic farmers. Agribusiness firms intensify their production by specializing on high-value crops. Small-scale producers have to adjust their farming practices in order to be able to compete with the large farms. Conventionalization leads to organic production as "organic lite" [17] (p. 301). The other scenario, the bifurcation hypothesis, states that only a few farms go through a process of conventionalization. These farms are large, professional, and strongly integrated in the market through contract farming with large supermarkets, as suppliers for large food industries, or as exporters. In this case, conventionalization processes are induced from the external environment, and only a small proportion of organic farms is affected [4]. Small farms are able to resist the conventionalization pressures by focusing on movement-based, locally oriented production [12]. Developments towards bifurcation were found in California [7], New Zealand [12], and Australia [3]. The bifurcation hypothesis could not be justified for the regions Ontario in Canada [5], West Germany [6], and Austria [4].

The debate on conventionalization has a number of general shortcomings [11]. The debate is criticized for its lack of conclusive data. Many researchers argue that more empirical studies are required before theories can be built [3,11,12,15]. Many studies are qualitative analyzes and do not include time-series data [11] or work with a restricted number of observations. Investigations that rely on quantifiable indicators of conventionalization measured over a certain time period are rarely found. Although such a research approach is limited by what can be measured and counted, its advantage is that we can investigate changes over time on complex issues [19]. None of the previously published conventionalization studies followed an approach combining different indicators to an aggregated conventionalization index. We consider our contribution as being an interesting complementary approach to qualitative conventionalization studies.

The objective of the paper is to develop composite indicators of conventionalization to measure the degree of conventionalization of single farms and of the whole organic farming sector. By means of the constructed composite indicators, we investigate whether conventionalization processes occur in the German organic farming sector.

Based on the principles of the organic movement, indicators to measure conventionalization are identified from bookkeeping data, which reflect the outcome of economic decisions of farm managers and are, therefore, able to capture potential developments towards conventionalization. The indicators are combined to composite indicators that are then applied to bookkeeping data of ten years. As a result, we can quantitatively assess processes towards conventionalization over time. Composite indicators or indices of conventionalization can serve as an assessment approach to verify the compliance of

organic farming practices with the principles of organic agriculture. They could be used, e.g., as a tool to monitor the developments of organic farming and to evaluate agricultural policies and measures [4,11]. Because organic quality mainly depends on the agricultural production process [11], this paper focuses on the investigation of developments on the farming sector, even though the debate of conventionalization spreads over the whole organic production chain.

In the following section, the approach of composite indicator construction is presented in five steps. A substantial part involves the identification of indicators that describe characteristics of conventionalization of organic farming (step 2). Section 3 presents and discusses the results of the composite indicators of German farms between 2000 and 2009. The paper concludes with a brief summary, along with the relevance and limitations of the main findings.

## 2. Materials and Methods

To develop an objective, purely quantitative assessment system of conventionalization is challenging since conventionalization is a complex phenomenon described by various indicators measured in different dimensions. A possibility to quantitatively investigate multi-dimensional processes is the combination of several indicators to a composite indicator. In this study, the construction of composite indicators of conventionalization was based on Gómez-Limón and Riesgo [20] and Gómez-Limón and Sanchez-Fernandez [21] who constructed composite indicators of agricultural sustainability. Their approach is in line with the steps to construct composite indicators presented in the user guide jointly prepared by the Organisation for Economic Co-operation and Development (OECD) and the Joint Research Centre of the European Commission (JRC) [22,23]. The construction of composite indicators involves decisions on data selection, data cleansing, normalization, weighting, and aggregation. These choices are in part subjective and influence the outcome of the constructed composite indicators [22–24]. The following steps were conducted in this study:

1. Development of the theoretical framework;
2. Indicator selection and data quality check;
3. Normalization;
4. Multivariate analysis;
5. Weighting and aggregation.

### 2.1. Step 1. Development of the Theoretical Framework

The presentation of the theoretical framework is the basis for the construction of a composite indicator. A clear definition of the phenomenon to be measured is the prerequisite to select and combine indicators in a meaningful way [23]. Conventionalization of organic agriculture is, however, not clearly defined. The debate about the validity of the conventionalization hypothesis is a result of different understandings of conventionalization. For a clear definition of conventionalization, we first need to clarify the understanding of organic agriculture. Since organic farming is based on values, De Wit and Verhoog [25] stated that only normative values are suitable as a guide to define organic agriculture. Therefore, the analysis of the phenomenon conventionalization is a normative one. A normative value basis of organic farming is provided by the principles of health, ecology, fairness, and care of the International Federation of Organic Agriculture Movements (IFOAM), which can be considered as the internationally shared values of the organic movement [11,25–28].

European Union (EU) legislation on organic agricultural production partly reflects the IFOAM principles [29]. Although the organic standards of the EU are approved by IFOAM, experts claim that the EU legislation leaves room for organic farming practices that can be in conflict with the core organic principles [8,14,18,25,30]. The regulatory focus on inputs would encourage farmers to simply substitute allowed inputs for disallowed inputs without considering the organic values [15,17,31,32].

Resulting from the definition of organic agriculture by IFOAM and the legislation of the EU, conventionalization is defined here as a process of changes of organic agriculture that are in accordance

with the organic regulations (of the EU) but in disagreement with the principles of organic agriculture (of IFOAM). If developments conform to the organic principles, they should be regarded as processes towards professionalization of organic farming.

### 2.2. Step 2. Indicator Selection and Data Quality Check

The indicators chosen in the investigation have to be relevant to the phenomenon being measured. The main problem of the conventionalization debate is to find suitable indicators that give clear evidence about conventionalization processes [8]. In order to identify which variables are appropriate in this context, Darnhofer et al. [11] proposed a set of indicators that relate to the IFOAM principles. The indicators were identified based on the approach of Sustainability Assessment of Farming and the Environment (SAFE) of van Cauwenberg et al. [33], which was also suggested by Gómez-Limón and Sanchez-Fernandez [21] as a guideline to select indicators. The SAFE framework of van Cauwenberg et al. [33] follows an holistic and hierarchical approach, such that, based on the principles of sustainability, criteria that capture the specific objectives of a farming system and the corresponding indicators are identified. As an example, from the principle "supply of quality water function" from the sustainability dimension "environment", the criterium "soil water of adequate quality is supplied" is derived. A corresponding indicator is the pH-value of soil water. A similar line of reasoning was followed by Darnhofer et al. [11] to identify indicators of conventionalization. Based on the four organic principles of IFOAM (health, ecology, fairness, and care), they identified indicators that point to conventional ways of thinking and conventional approaches to solve a problem, such as too high production targets that are not adapted to the location, lack of understanding of organic interactions, and imbalances between short- and long-term goals. As an example, the indicator "high share of cereals in crop rotation" is based on the principle of ecology, because a high share of cereals in crop rotation can lead to plant diseases. It is, therefore, in conflict with the objective, i.e., criterium, of healthy plants. However, most of the indicators that Darnhofer et al. [11] presented are very specific. They can only be verified by means of personal interviews and field studies. As a consequence, we only choose the indicators suggested by Darnhofer et al. [11] that can be straightforwardly calculated through information provided by our dataset (as proposed by Gómez-Limón and Sanchez-Fernandez [21]).

The data used in this study were from the Farm Accountancy Data Network (FADN) [34]. FADN is a sample survey system that collects structural and accountancy data of agricultural holdings in the member states of the European Union.

FADN is the only data set harmonized through bookkeeping principles applied in all member states. Organic farming is included in the FADN dataset since 2000. In 2000, 151 organic farms (26 converted organic farms and 125 farms in conversion) out of 6233 farms in total (2.4%) were included in the FADN dataset of Germany. In 2009, 403 organic farms (26 converted farms and 377 in conversion) were recorded, i.e., 4.6% of the 8812 farms in the full sample. In this investigation, no distinction is made between farms in conversion and already converted farms.

Based on the proposed indicators of Darnhofer et al. [11] and with reference to the best practice guideline of organic farming of IFOAM representing approaches and methods in accordance with the organic principles published by the the Sustainable Organic Agriculture Action Network (SOAAN) [35], eight conventionalization indicators were identified from Darnhofer et al. (2010):

- High share of cereals in the crop rotation;
- Reliance on easily soluble (nitrogen) fertilizers (e.g., vinasse);
- Prolonged and intensive use of plant protection products that are known to be problematic (e.g., sulfur, copper, pyrethrum);
- Widespread use of practices that require a high level of external inputs (energy, fertilizers, feedstuffs, materials);
- Low level of biodiversity on the cropland;

- High incidence rate of metabolic disorders/frequent veterinary treatments required, high incidence rate of abnormal animal behavior;
- High share of feed that is purchased (industrially produced) rather than produced on the farm (or by neighbouring farms);
- Low time budget for the care of the individual animal and for herd management [11] (pp. 76–78).

Many of the chosen indicators indicate the intensification of organic farming. This is reasonable because most intensification processes (reliance on high-performance breeds and species, as well as increases in purchased inputs) are in conflict with the organic principles [8,17]. All identified and constructed conventionalization indicators denote a stronger degree of conventionalization the larger their values get.

### 2.2.1. Fertilizer and Crop Protection per ha (FERTIL, CROPPROT)

The use of fertilizer and plant protection products should be avoided in organic farming. However, some products are allowed by the EU. The regulation (EC) No 889/2008 of the European Commission states that organic farmers should significantly restrict the use of pesticides and, if applied, record the reason for the use of fertilizer and plant protection [36]. The European Commission provides a catalog with allowed crop protection products for organic farming in annex II of the regulation (EC) No 889/2008. A strong reliance on fertilizer and crop protection products is seen as a threat to the principles of ecology and health as it is often used as substitute for a more diverse rotation [4,11,18,25]. The variable reflects the IFOAM principle of ecology, teaching organic farmers to prevent water pollution caused by the extensive use of fertilizers or other chemicals [35]. The use of fertilizer and crop protection products are, therefore, often considered as indicating conventionalization processes [4,11,18,25].

### 2.2.2. Cereal Share (CERESHAR)

Darnhofer et al. [11] argued that a high share of cereals in crop rotation leads to deficiencies of nitrogen supply and humus build-up. They argued that an increase in cereal growing is in conflict with the principle of ecology. The best practice guideline of IFOAM says that building soil of high quality is important for the sustainability of the agricultural system [35]. According to Groier [4], a high share of cereals is a process of specialization of organic farming causing an increase in the degree of conventionalization. To calculate the share of cereals, the area upon which cereals grow was divided by the total utilized agricultural area.

### 2.2.3. Diversity of Crops Planted (CROPDIV)

The diversity of crops planted is an essential element of organic farming. A reduction is considered as a step towards conventionalization [4,11]. Diversity in crop production should reflect the diversity of nature, according to SOAAN [35]. Therefore, organic farmers are encouraged to increase the number of crop species planted [35]. The indicator was constructed based on the Shannon index which is a measure of species diversity. The Shannon index was widely used to calculate the evenness and richness of species in a vegetal or animal assembly [37,38]. Sipiläinen and Huhtala [37] and Pacini et al. [38] applied the Shannon index to cultivated rather than wild species. Instead of considering the number of species in the calculation, crop acreages were used. The higher the Shannon Index is, the higher crop diversity is [37]. Since the other utilized indicators in this research are positively related to conventionalization, the indicator is calculated as one minus the Shannon Index. Thus, a high indicator value points to low crop diversity and, with it, stronger conventionalization.

### 2.2.4. Feed Purchase per Livestock Unit (LU) (FEEDPURC)

Organic farming should foster regenerative agricultural systems that rely on closed nutrient cycles [35]. Strong reliance on external feeding stuff is linked to an unbalanced energy use on organic farms, which is in conflict with the principle of ecology [4,11]. Also, IFOAM states that, ideally, the

farms should only raise the amount of animals they can feed [35]. To take the herd size into account, the sum of purchased feed for grazing livestock and feed for pigs and poultry is divided by the total number of livestock units (LU). The smaller the ratio is, the closer the farm acts in accordance with the principle of ecology. A value of zero is considered as optimal as the farm does not rely on any external feed. The type of feed, whether it contains natural feed or synthetized components, might have an even stronger ecological impact than the origin of feed. However, as our database did not include information about the type of feed, it was not possible to use this type of information within our model.

### 2.2.5. Herd Size in Relation to Time Budget (HERDSIZE)

Farms should manage the herd size well. IFOAM teaches organic farmers to give attention to and to interact with the animals, which is the requirement for a proper consideration of their intrinsic value [35]. The number of livestock units in relation to total labor unit offers information about the time budget available for the care of the individual animal and herd management. An increase in herd size reduces, ceteris paribus, the time budget per animal. Low time budget for animals causes animal welfare to suffer [11]. Also, De Wit and Verhoog [25] argued that the value of animal health and welfare is inconsistent with large-scale, specialized pig and poultry units in which the number of human-livestock interactions is reduced. This would be in contradiction to the principles of health, fairness, and ecology [4,11,25]. The division of total livestock units by total labor input—corrected for the time spent on cropping activities based on standard labor input coefficients from the standard technical handbook on agricultural production activities in Germany [39]—gives the number of animals that could be cared for in one working hour. An increase of the ratio means that one worker has to care for more animals, i.e., they have less time for the individual animal. This indicator is only a proxy, since the FADN variable "total labor input" is based on estimates of the farmers. It entails, therefore, a component of uncertainty.

### 2.2.6. Stocking Density (STOCDENS)

As Darnhofer et al. [11] stated, atypical animal behavior can be due to a lack of space to allow natural behavior. A high degree of animal production density means an intensification of animal husbandry farming [4,18]. Farms should appropriately manage stocking density to avoid animals harming each other. This reduces the incidence of diseases and environmental pollution as IFOAM states in its guideline [35]. According to the principle of fairness, each species should be able to exhibit typical behavior, such as grazing for cows. Therefore, a high stocking density does not respect the principles of fairness and ecology [4,11,35]. The variable stocking density was provided by FADN.

### 2.2.7. Livestock-Specific Costs per LU (ANIMCOST)

This indicator is a proxy for the incidence rate of metabolic disorders and veterinary treatments. A high amount of expenditures for veterinary treatments is in conflict with the principle of health [11]. The farms should avoid disorders by preventative health measures. Natural remedies should be taken instead of chemical treatments except of emergency cases [35]. The FADN variable "other livestock-specific costs" includes veterinary fees and reproduction costs which are expected to increase with intensified production towards conventionalization [4]. A high share of reproduction through artificial insemination is considered as an indicator of conventionalization [11]. Darnhofer et al. [11] argued that animals should be allowed to mate naturally (principle of fairness). The FADN variable further includes costs for milk tests, occasional purchases of animal products (milk, etc.), and costs incurred in the market preparation, storage, marketing of livestock products, etc. The variable is, therefore, no clear indicator of processes that are in conflict with the organic principles.

The whole set of indicators is only applicable to farms that are active in both crop production and animal husbandry. Farms without animals were evaluated on the basis of the four indicators that are relevant for organic crop production: FERTIL, CROPPROT, CROPDIV, and CERESHAR. For the farms that are specialized on animal husbandry, the six indicators FERTIL, CROPPROT,

FEEDPURC, HERDSIZE, STOCDENS, and ANIMCOST were of interest, which resulted in a more complete description of mixed farms compared to the specialized farms. Many specialized animal farms in our dataset use crop protection products on their arable land on which they grow feed. That is why we included this variable in the analysis of this farm type.

Since the ability of a composite indicator to represent multidimensional phenomena depends on the quality and accuracy of its components [22], the dataset was checked for outliers, data gaps, and plausibility, resulting in a dataset reduced by two to eight farms per year (Table 1). Furthermore, the five indicators that are measured in euros (FERTIL, CROPPROT, FEEDPURC, ANIMCOST) were deflated with the corresponding agricultural price indices of the German federal bureau of statistics (2010 = 1) [40].

**Table 1.** Number of farms analyzed.

| Year | 2000 | 2001 | 2002 | 2003 | 2004 | 2005 | 2006 | 2007 | 2008 | 2009 |
|---|---|---|---|---|---|---|---|---|---|---|
| **No. of organic farms in the FADN dataset** | 151 | 256 | 296 | 291 | 306 | 298 | 317 | 323 | 354 | 403 |
| **Total no. of farms included in analysis** | 147 | 253 | 289 | 288 | 298 | 293 | 315 | 316 | 346 | 395 |
| **No. of crop farms** | 11 | 35 | 41 | 42 | 46 | 49 | 53 | 51 | 54 | 65 |
| **No. of animal farms** | 24 | 45 | 55 | 62 | 57 | 54 | 55 | 58 | 65 | 72 |
| **No. of mixed farms** | 112 | 173 | 193 | 184 | 195 | 190 | 207 | 207 | 227 | 258 |

**Source**: Own calculations, based on FADN data [41]. **Note**: FADN—Farm Accountancy Data Network.

### 2.3. Step 3. Normalization

Normalization is necessary to make the data comparable because the different indicators do not have a common meaningful unit of measurement and differ in their range [23,41,42]. For any aggregation and weighting method, the effective weight of the indicators depends on the measurement units and on the range. Therefore, the choice of normalization has an influence on the overall outcome [43]. In this paper, min–max–normalization (or re-scaling) was applied. The normalized indicators have a common range between 0 and 1, with 0 as optimal value and 1 as the worst value indicating a high degree of conventionalization. Since the methods of multivariate analysis require a normalized dataset, normalization was conducted prior to multivariate analysis [22,44].

### 2.4. Step 4. Multivariate Analysis

To look at the single indicators as own entities is an indispensable step towards the construction of a composite indicator [22,23]. Multivariate analysis consists of a set of statistical tests that provide insight into the overall structure of the indicators, the suitability of the data set, and the methodological choices on how to proceed during the next steps [21,22]. Information in the dataset was grouped along two dimensions. One dimension was the range of the indicators; the other dimension was represented by the different farms [22]. Information on farms was analyzed with cluster analysis. The results indicated that a division of the dataset into subgroups according to three different farming types was appropriate. Principal component analysis (PCA), which is applied to group information of indicators, was, therefore, applied for the three subgroups: farms specialized on crop production, farms specialized on animal husbandry, and mixed farms.

### 2.5. Step 5. Weighting and Aggregation

Weighting and aggregation of indicators are the central steps to combine the indicators to a single index in a meaningful way. The choice of the aggregation procedures comes together with the choice regarding the weighting of indicators [22,45]. We derived our weights from PCA and used linear aggregation.

The idea of PCA is to reduce the dimensionality of a dataset that consists of a large number of correlated variables and to retain as much as possible of the variation of the dataset [46]. The correlated variables were transformed into a smaller number of uncorrelated factors, the principal components (PC) [47]. The principal components are linear combinations of the original variables. Their coefficients are called loadings; they indicate to what extent the specific indicator accounts for the prevailing principal component. The principal components capture different amounts of variance of the original variable set. The first principal component captures most of the variation in all of the original variables, the second principal component captures most of the remaining variation, and so on [22,46]. The weighting using PCA is endogenous and represents the statistical importance of the individual indicators [22].

Nardo et al. [22] and Nicoletti, Scarpetta, and Boylaud [48] presented a method of combined weighting by means of PCA and linear aggregation, which was applied by Gómez-Limón and Sanchez-Fernandez [21] and Gómez-Limón and Riesgo [20]. The procedure consists of the following six steps:

1. Calculation of correlation matrix;
2. Identification of number of principal components necessary to represent the dataset;
3. Rotation of weights;
4. Construction of weights;
5. Construction of intermediate composite indicator;
6. Aggregation of intermediate composite indicators to final composite indicator.

We firstly checked the correlation between indicators. If the indicators are uncorrelated, a principal component analysis would not be appropriate as it is based on correlations [48]. At the same time, the correlations should not be too high to make sure that the indicators do not all measure the same development [22,46]. Afterwards, we used the results of PCA from step 4 to identify how many principal components were necessary to represent the variance in the dataset. A frequent practice is to choose the principal components that have an eigenvalue close to or larger than one, individually represent at least 10% of the overall variance, and cumulatively contribute to the explanation of the total variance by at least 60% [22]. Table 2 exemplarly represents the principal components of crop farms. We kept the first two principal components.

**Table 2.** Eigenvalues of the indicator set of crop farms. PC—principal component.

|  | Eigenvalue | Proportion of Variance | Cumulative Proportion |
|---|---|---|---|
| **PC 1** | 1.755 | 43.883 | 43.884 |
| **PC 2** | 1.000 | 25.011 | 68.895 |
| **PC 3** | 0.791 | 19.777 | 88.671 |
| **PC 4** | 0.453 | 11.329 | 100.000 |

**Source**: Own calculations, using the R-package "FactoMineR" of Lê et al. [49].

The selected principal components were rotated in order to get a clear pattern of loadings and a simpler structure of the principal components. Following Nicoletti et al. [48], Nardo et al. [22], and Gómez-Limón and Riesgo [20], we applied the varimax rotation, which minimizes the number of variables that have high loadings on a principal component. We kept the rotated loadings that were greater than 0.3 for the further construction steps [22,48]. For the group of crop farms, we deleted the rotated loadings of the second principal component for the indicators FERTIL and CROPPROT (Table 3).

**Table 3.** Rotated loadings for crop farms. FERTIL—fertilizer per ha; CROPPROT—crop protection per ha; CROPDIV—diversity of crops planted; CERESHAR—cereal share.

|  | PC 1 | PC 2 |
|---|---|---|
| FERTIL | 0.587 | −0.225 |
| CROPPROT | 0.847 | 0.101 |
| CROPDIV | 0.821 | 0 |
| CERESHAR | 0 | 0.978 |

**Source**: Own calculations, using the R-package "FactoMineR" [49].

The construction of weights is based on the two following formulae:

$$w_{kj} = \frac{\left(rotated\ factor\ loading_{kj}\right)^2}{eigenvalue_j},$$ (1)

$$\alpha_j = \frac{eigenvalue_j}{\sum_j eigenvalue_j},$$ (2)

where rotated factor loading *kj* is the value of the loading of indicator *k* in the principal component *j*, and eigenvalue *j* is the eigenvalue of the *j*th principal component [20].

Equation (1) gave a weighting matrix which included the individual weights of the indicators in the principal components. Each indicator was weighted according to the proportion of its variance that was explained by the principal component it was associated with. Each principal component was then weighted according to its contribution to the explained variance in the dataset using Equation (2) (see Tables 4–6).

**Table 4.** Weights of crop farms.

|  | PC 1 | PC 2 |
|---|---|---|
| **Weighting matrix of indicators ($w_{kj}$)** | | |
| FERTIL | 0.196 | 0 |
| CROPPROT | 0.409 | 0 |
| CROPDIV | 0.384 | 0 |
| CERESHAR | 0 | 0.956 |
| **Weights of PC ($\alpha_j$)** | **0.637** | **0.363** |

**Source**: Own calculations.

**Table 5.** Weights of animal farms. FEEDPURC—feed purchase per livestock unit (LU); HERDSIZE—herd size in relation to time budget; STOCDENS—stocking density; ANIMCOST—livestock-specific costs per LU.

|  | PC 1 | PC 2 | PC 3 |
|---|---|---|---|
| **Weighting matrix of indicators ($w_{kj}$)** | | | |
| FERTIL | 0 | 0.500 | 0 |
| CROPPROT | 0 | 0.500 | 0 |
| FEEDPURC | 0.303 | 0 | 0.108 |
| HERDSIZE | 0 | 0 | 0.613 |
| STOCDENS | 0.419 | 0 | 0 |
| ANIMCOST | 0 | 0 | 0.800 |
| **Weights of PC ($\alpha_j$)** | **0.454** | **0.331** | **0.216** |

**Source**: Own calculations.

**Table 6.** Weights of mixed farms.

|  | PC 1 | PC 2 | PC 3 | PC 4 |
|---|---|---|---|---|
| **Weighting matrix of indicators ($w_{kj}$)** | | | | |
| FERTIL | 0 | 0.490 | 0 | 0 |
| CROPPROT | 0 | 0.468 | 0 | 0 |
| CROPDIV | 0.440 | 0 | 0 | 0 |
| CERESHAR | 0.440 | 0 | 0 | 0 |
| FEEDPURC | 0.081 | 0 | 0.242 | 0.143 |
| HERDSIZE | 0 | 0 | 0.314 | 0 |
| STOCDENS | 0 | 0 | 0 | 0.831 |
| ANIMCOST | 0 | 0 | 0.421 | 0 |
| **Weights of PC ($\alpha_j$)** | 0.341 | 0.271 | 0.207 | 0.181 |

**Source**: Own calculations.

Afterwards, the weighted indicators were combined to an intermediate composite indicator. Nicoletti et al. [22] foresees to group the indicators to an intermediate composite indicator using the weights $w_{kj}$ from Equation (1).

$$ICI_{ji} = \sum_{k=1}^{n} w_{kj}I_{ki},\tag{3}$$

where $ICI_{ji}$ is the intermediate composite indicator for the principal component $j$ and the farm $i$, $w_{kj}$ is the weight of indicator $k$ in the component $j$, and $I_{ki}$ is the normalized indicator $k$ achieved by farm $i$. The intermediate indicators were then aggregated to the final composite indicator using the weights $\alpha_j$ derived from Equation (2).

$$CI_i = \sum_{j} \alpha_j ICI_{ji},\tag{4}$$

where *CIi* is the value of the composite indicator for the farm $i$ [20].

We applied this methodology to the dataset of the observation year 2009. Thus, the weights of the resulting composite indicators were based on organic farms reported in 2009. The weights were then applied to the period from 2000 to 2008. The reason for this approach was that the sample size in 2009 was the largest within the 10 years under investigation and, therefore, one can expect a higher reliability.

## 3. Results and Discussion

The results of PCA of the three subgroups suggested that, for the farms specialized on crop production, we only kept the first two principal components (PC) for the construction of composite indicators, as presented above. In the case of the specialized animal farms, we kept the first three principal components, and, for the mixed farms, the first four principal components were selected. After varimax rotation, we calculated the weights of indicators and principal components, presented in Tables 4–6, according to Equations (1) and (2). Some indicators were assigned to a high weight, such as CERESHAR with 0.956 in the case of crop farms (Table 4). However, since the indicator variable CERESHAR was assigned to the second principal component (PC 2), which had a relatively low weight in the composite indicator ($\alpha_2 = 0.363$), CERESHAR did not have a considerably stronger influence than other indicators. Similar findings could be observed for the other subgroups. Therefore, none of the variables had a dominant influence on the results of the composite indicators.

Linear aggregation of the weighted indicators and principal components according to Equations (3) and (4) gave the composite indicators of crop farms (CI.crop), animal farms (CI.anim), and mixed farms (CI.mixed). The developments of the means are presented in Figure 1. The scores of the composite indicator of crop farms are higher than the scores of the mixed and the animal farm composite indicators. This may indicate that the farms specialized on crop production show a higher degree of conventionalization than the other farm types. General developments towards more

conventionalization from 2000 to 2009 could not be observed within the three subgroups. We also investigate the fifth and 95th percentiles of the three subgroups in order to detect potential developments towards increasing dispersion of the degree of conventionalization among the farms. However, the developments of the percentiles do not show such a pattern.

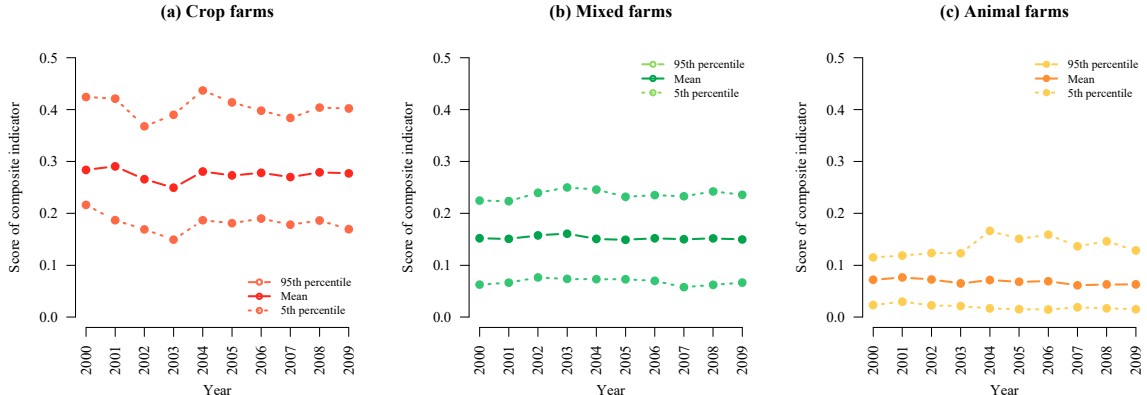

**Figure 1.** Development of means and percentiles of composite indicators (CIs) for the three farm types: (**a**) Crop farms; (**b**) Mixed farms; (**c**) Animal farms (source: own calculations).

The distribution of the composite indicators for the years 2001, 2005, and 2009 are presented in Figure 2a–c, which give more information about the distributional shape and changes over time. The distributions of the composite indicators of crop farms (Figure 2a) and of animal farms (Figure 2c) are right-skewed. The right-skewness indicates that there are some farms with a higher degree of conventionalization. The degrees of skewness of the years 2001 and 2009 for the subgroups crop, animal, and mixed farms do not differ significantly from each other. The distributions of the two specialized farm groups show slight local maxima on the righthand side of the graph. These maxima give support for the bifurcation hypothesis. There seems to be a small number of organic farms that show relatively high degrees of conventionalization.

Bifurcation seems to be an issue in the German organic farming sector. This conclusion is in contrast to the findings of Best [6], who did not find evidence for bifurcation in West Germany. The study of Best [6] was based on a questionnaire in which attitudes of farmers towards, e.g., the organic movement were recorded. However, farmers have to balance out their organic attitudes with economic interest. As a result, farming decisions do not necessarilty have to fit to the attitudes of organic farmers. Our composite indicators capture this decision-making of organic farmers over a time period of 10 years and can, therefore, present results to be observed on the market.

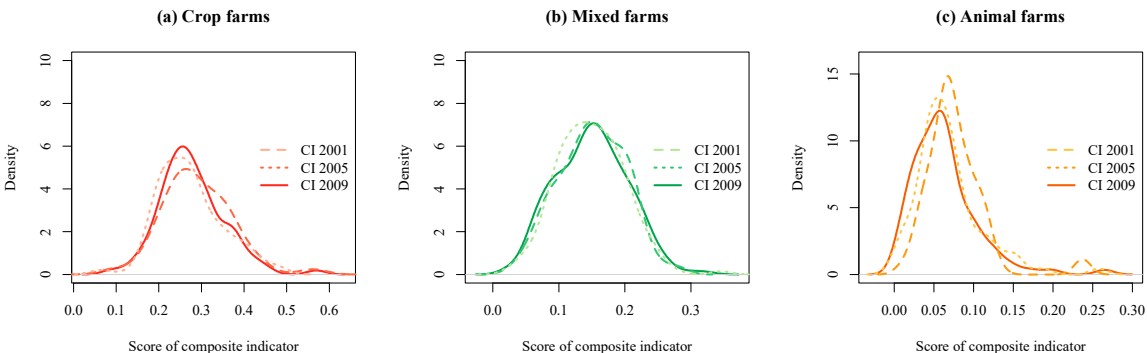

**Figure 2.** Distribution of the composite indicators for 2001, 2005, and 2009: (**a**) Crop farms; (**b**) Mixed farms; (**c**) Animal farms (source: own calculations).

The constructed composite indicators also allows the investigation of the development of single farms. Figure 3 depicts the developments of randomly selected farms included in the FADN dataset

from 2000 (2001 for crop farms) to 2009. The figures show that there were no uniform developments among the farms. The composite indicators of three out of the four depicted crop farms (Figure 3a) are increasing during the past two to four years. The composite indicator score of the fourth farm, however, is strongly fluctuating without a clear tendency towards conventionalization. Also, the developments of the composite indicators of four animal farms are inhomogeneous (Figure 3c). Of particular interest is the development of the mixed farms (Figure 3b). While farms 2 and 3 moved in the area of the mean, which is between 0.14 and 0.15, farm 1 shows a strong increase in the degree of conventionalization, reaching a composite indicator score in 2009 of 0.26. Farm 4 develops in a different direction with a decreasing composite indicator score. This finding suggests that farm 1 is one of the farms that are part of the small local maxima of the density function. Hence, although there is no general development towards conventionalization, individual farms can show a clear increase in the degree of conventionalization, again supporting the bifurcation hypothesis on the single-farm level.

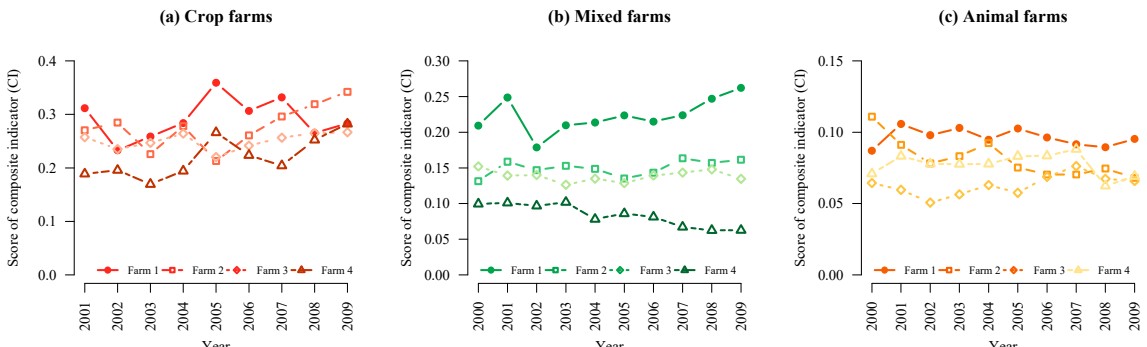

**Figure 3.** Development of composite indicators (CIs) of single farms for the three farm types: (**a**) Crop farms; (**b**) Mixed farms; (**c**) Animal farms (source: own calculations).

In general, due to our findings of bifurcation on the single farm level, we can partly support the findings of Oppermann [1] and Best [6], who found slight tendencies towards conventionalization in Germany. A general strong development on a sectoral level towards highly conventionalized farms, as found in California by Buck et al. [7], could not be found for the German organic sector.

## 4. Conclusions

The objective of this paper is to empirically investigate the phenomenon of conventionalization of organic farms in Germany through the construction of a composite indicator. The research question is whether conventionalization can be detected by composite indicators.

Conventionalization of organic agriculture was theoretically debated since the late 1990s. The term conventionalization was roughly defined within the context of sociological studies. However, diverging concepts led to a confusion of conventionalization with processes of professionalization, which can also be regarded as normal developments in a changing economic environment. Furthermore, there were little attempts in the literature to quantify the process of conventionalization.

This study takes a systematic approach in that regard. Following several suggestions in the literature, conventionalization is defined here as processes that undermine the organic principles despite being in accordance with the EU legislation of organic agricultural production. Using bookkeeping data from FADN between 2000 and 2009, eight indicators were identified to give evidence about conventionalization processes on farms. Since not all indicators could be calculated for both crop production or on animal husbandry, the dataset was divided, and the construction of acomposite indicator was done separately for farms that specialized in either one, along with those being active in both farming activities.

The main findings of the study allowed a number of conclusions as follows:

(1) The method is able to depict and quantify the process of conventionalization based on a large data sample. Bookkeeping data are the result of economic decisions of organic farmers and reflect relevant decisions of the actors. Therefore, they can be used to model the process of conventionalization.

(2) The quantitative results show an individual process of conventionalization at the farm level. There seems to be a small number of farms that moved towards conventionalization, as shown by composite indicator scores of single farms. The results suggest that some decision-makers moved away from IFOAM organic principles of health, ecology, fairness, and care, using, e.g., large crop rotations instead of means-based crop protection or improved animal husbandry instead of last-minute treatment of single animals. However, these implications have to be considered with care, since high expenses, e.g., for animal treatment, might be caused by single events. Nevertheless, in some cases, the indices increased over many years, which highlights the importance and potential to model conventionalization based on time-series and panel data.

(3) From a sector's perspective, the results do not verify a large-scale process of conventionalization for *all* farms. The constructed composite indicators do not give clear evidence of a general process of conventionalization on German farms between 2000 and 2009. The average degree of conventionalization of the three subgroups do not show a definite increase during this time period. However, some slight local maxima of the indicator density at higher composite indicator scores might indicate that a small number of farmers have a higher degree of conventionalization.

(4) These two findings suggest a bifurcation of the organic farms into a small number of large, relatively highly conventionalized farms on the one hand, and a larger number of small, non-conventionalized farms on the other.

With respect to the limitations, it must be stressed that the presented developments do not detail the diverse processes occurring at the farm level. They rather give information regarding a small part of organic farming and processes of conventionalization, the part that is captured by the indicators. Structural change, as well as changing market conditions and consumer demands, may also cause an increase in the different indicators. We argue that while the single variables might not unambiguously indicate conventionalization processes, the whole set of indicators gives evidence of conventionalization. Moreover, whether a purely quantitative investigation is able to capture heterogeneous developments of organic agriculture to clearly depict the normative concept of conventionalization may remain an issue of general discussion. However, if the debated phenomenon of conventionalization was strong, we would expect there to be clear and measurable evidence at the farm level for it.

We would like to note, in this context, that our definition of conventionalization depends on the set of organic values and standards. Parts of the conventionalization debate are related to perceptions of the organic market by actors. The proposed model is not able to capture attitudes and expectations of farm managers. It might be an interesting challenge, for future research, to develop a farm-specific indicator using both input variables (decision outcomes) and perception or expectation measures. Consequently, the question whether organic farming is subject to the process of conventionalization is strongly linked to the expectations of farmers, consumers, and citizens towards the organic farming system underlying the definition of organic standards. Therefore, conclusions from conventionalization are a political issue, to be discussed by organic farming associations and other relevant actors in the field.

This study presents a possible approach to empirically investigate conventionalization processes at farm level that can be further developed to obtain an easily understandable tool simplifying and summarizing the complex and multivariate development of organic farming. Composite indicators that assess conventionalization on organic farms could be used by policy-makers as a reference to monitor the development of organic agriculture or as a basis for evaluations and the further development of

organic regulations and measures. With such a tool, we are able to find evidence of the relevance of conventionalization of organic farming in different regions. If detected, an undesirable development of organic farming could induce a targeted discussion whether regulation is needed to counteract the processes and what type of focus in the regulation is warranted. Regarding our specific results on organic farming practices in Germany between 2000 and 2009, we do not find that organic farming in Germany is generally threatened by conventionalization, even if this might be different for single farms. Thus, we do not see the need for changes in regulation of the organic sector going forward.

**Author Contributions:** Conceptualization, C.S., T.H. and S.L.; Formal analysis, C.S. and S.L.; Investigation, C.S.; Methodology, C.S.; Supervision, T.H. and S.L.; Writing—original draft, C.S.; Writing—review & editing, C.S., T.H. and S.L.

**Funding:** This research received no external funding.

**Conflicts of Interest:** The authors declare no conflicts of interest.

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
