# Peer review of "Conventionalization of Organic Farms in Germany: An Empirical Investigation Based on a Composite Indicator Approach"

_sustainability, doi:10.3390/su11102934_

Reviewer 1 Report

This paper is very interesting and follows an emerging theme framed on the Sustainability of the agriculture sector. It is well organized too.

However, some revisions are needed:

(1) Objective: The main objective of the paper needs clarification in two ways. 

The first part is vague “The objective of the paper is to quantitatively investigate potential conventionalization processes…” (line 84) and needs to be clarified and disassembled.

The second part “…in the German organic farming sector by constructing composite indicators of conventionalization” (lines 85) has to adequately detail the used dimension(s) – environmental, economic, social, other - for the measure of the degree of conventionalization. This will orientate the selection of the indicators to integrate in the composite indicator of conventionalization and the analysis of the results.

(2) State of the art: The theoretical framework exposed in introduction is relevant and well done, but it is needed more detail regarding the not clearly concept of “conventionalization of organic agriculture”. The authors advanced with an own concept, but it is needed to substantiate it “conventionalization of organic agriculture is a process of changes of organic agriculture that are in accordance with the organic regulations (of the EU) but in disagreement with the principles of organic agriculture (of IFOAM).” The references to this concept are developed in different sections (1. Introduction; step 1 of section 2; and section 3), but I suppose that the paper could improve if they were together.

Also there are other frontier concepts, as e.g. the “Sustainable intensive farming”, that can be related with the thematic, if the authors want.

(3) Methodology: The authors used similarities with SAFE approach, but not all dimensions of sustainability are covered in the selected indicators used for the paper (question related with the objectives).

The studies of Gómez-Limón and Riesgo [20] and Gómez-Limón and Sanchez-Fernandez [21] were cited, but it is necessary more information regarding the composite indicators that they used and their results for a better justification for this paper.

The indicator “Feed purchase per LU (FEEDPURC)” is a sustainability indicator, relevant to the autonomy of the farm and the system, but regarding the ecological principles the type of feed (natural feed or a synthetized compounds) has a more pronounced ecological impact.

(4) Results and discussion: It is very complex to follow the steps of the proposed procedure (lines 187-189) without the exhibition of their results. The used data for that and/or that results needs to be presented, even if it is done by a briefly way or in an appendix. 

In addition, it should be noted that the results exposed in the paper were not discussed with other previous works, and it is necessary that discussion.

(5) Some mistakes found:

- line 151 has missing the end of the parenthesis “[33,34,23”

- line 218 has missing “is” (“where ICIij “is” …);

- line 230 “3. Indicator selction” should be correted for “3. Indicator selection”

- lines 244 and 245 have to be together;

- line 267 the reference has to be numbered “(Darnhofer et al., 2010, p. 76ff).

- line 364 the reference has to be numbered (FADN data).

Author Response

Dear Reviewer,

thank you very much for your review that helped us a lot to improve our paper. In the following, we will present our reply to your points of criticism referring to the lines of the reviewed paper under “track changes” mode.

Kind regards

The Authors

 (1) Objective: The main objective of the paper needs clarification in two ways.

The first part is vague “The objective of the paper is to quantitatively investigate potential conventionalization processes…” (line 84) and needs to be clarified and disassembled.

The second part “…in the German organic farming sector by constructing composite indicators of conventionalization” (lines 85) has to adequately detail the used dimension(s) – environmental, economic, social, other - for the measure of the degree of conventionalization. This will orientate the selection of the indicators to integrate in the composite indicator of conventionalization and the analysis of the results.

Thank you for your remark. In order to make our study and our approach more comprehensible, we reformulated this section and described our research objective and approach more extensively. Please note that our study is about conventionalization of organic farming. Even though organic farming contributes to sustainable agriculture, the concept of conventionalization is not linked to the three sustainability dimensions. It is rather based on the four principles of the organic movement provided by IFOAM. As a result, our investigation of conventionalization is based on derivations of farming practices from the organic principles and not based on the three dimensions of sustainability. We apologize for not having stated this more precisely and hope that our reformulations clear up any misunderstandings.

(2) State of the art: The theoretical framework exposed in introduction is relevant and well done, but it is needed more detail regarding the not clearly concept of “conventionalization of organic agriculture”. The authors advanced with an own concept, but it is needed to substantiate it “conventionalization of organic agriculture is a process of changes of organic agriculture that are in accordance with the organic regulations (of the EU) but in disagreement with the principles of organic agriculture (of IFOAM).” The references to this concept are developed in different sections (1. Introduction; step 1 of section 2; and section 3), but I suppose that the paper could improve if they were together.

We agree with you that the integration of indicator selection in the second chapter, in which the approach is presented, enhances the comprehensibility of our study. Therefore, we fully integrated chapter 3 into chapter 2, step 2. Furthermore, we shifted the presentation of our definition of conventionalization from chapter 1 to chapter 2, in order to improve readability.

Also, there are other frontier concepts, as e.g. the “Sustainable intensive farming”, that can be related with the thematic, if the authors want.

Thank you for your remark. However, we did not include this or other frontier concepts in the paper in order not to provoke confusions of the conventionalization debate with sustainable intensive farming concepts.

(3) Methodology: The authors used similarities with SAFE approach, but not all dimensions of sustainability are covered in the selected indicators used for the paper (question related with the objectives).

As already described in the reaction to your first point, we tried to make the objective and methodology of our paper clearer by reformulating the introduction. Furthermore, we added information about the SAFE approach (l. 162 ff). In our study, we use conventionalization indicators of Darnhofer et al. [11]. Darnhofer et al. [11] identify conventionalization indicators based on the holistic and hierarchical approach of SAFE. However, the indicators are not sustainability indicators based on the three dimensions, but conventionalization indicators based on the organic principles. We hope that our additional description of the SAFE approach and the approach of Darnhofer et al. [11] make this differentiation clearer.

The studies of Gómez-Limón and Riesgo [20] and Gómez-Limón and Sanchez-Fernandez [21] were cited, but it is necessary more information regarding the composite indicators that they used and their results for a better justification for this paper.

Our conventionalization study is not directly linked to the sustainability studies of Gómez-Limón and Riesgo [20] and Gómez-Limón and Sanchez-Fernandez [21] from a topic’s perspective. We follow their methodological approach, but we do not have the same research topic, even though we acknowledge that organic farming is linked to the concept of sustainability. We fear that the presentation of the results of Gómez-Limón and Riesgo [20] and Gómez-Limón and Sanchez-Fernandez [21] might lead to confusions of the sustainability concept with the concept of conventionalization of organic agriculture.

The indicator “Feed purchase per LU (FEEDPURC)” is a sustainability indicator, relevant to the autonomy of the farm and the system, but regarding the ecological principles the type of feed (natural feed or a synthetized compounds) has a more pronounced ecological impact.

Thank you for this note. We agree with you. Unfortunately, we cannot include your proposed indicator as it is not included in our FADN-dataset.

4) Results and discussion: It is very complex to follow the steps of the proposed procedure (lines 187-189) without the exhibition of their results. The used data for that and/or that results needs to be presented, even if it is done by a briefly way or in an appendix.

In order to improve the comprehensibility of the construction of the composite indicators, we integrated intermediate results of the construction steps for the farm group of crop farms. We present the eigenvalues of principal components (table 2, l. 364) and the rotated factor loadings (table 3, l. 373) that are partly deleted and further processed to construct composite indicators. The weights for the three farm groups (crop, animal and mixed farms, l. 385 ff) and the formula 3 and 4 to construct the composite indicators are now placed directly one behind the other. This should further increase the comprehensibility of the approach of composite indicator construction.

In addition, it should be noted that the results exposed in the paper were not discussed with other previous works, and it is necessary that discussion.

Thank you for your recommendation. We extended the chapter by discussion parts referring to previous studies of conventionalization (l. 589 ff and 621 ff).

 (5) Some mistakes found:

- line 151 has missing the end of the parenthesis “[33,34,23”

- line 218 has missing “is” (“where ICIij “is” …);

- line 230 “3. Indicator selction” should be correted for “3. Indicator selection”

- lines 244 and 245 have to be together;

- line 267 the reference has to be numbered “(Darnhofer et al., 2010, p. 76ff).

- line 364 the reference has to be numbered (FADN data).

Thank you very much for your careful reading We corrected the mistakes.

Reviewer 2 Report

The article is focused on assessing the conventionalization of organic farming in Germany by creating composite indicators. The authors of the study used the construction of composite convention indicators based on a methodology consisting of indicators of agriculture sustainability (according to the OECD and JRCEC manuals). Thus, the study proceeds according to certain steps where the authors focus on the development of the theoretical framework, selection of the indicator, normalization, multivariate analysis and aggregation. The input values were therefore three types of farms, namely farms specialized on crop production, farms specializing on animal husbandry and mixed farms. Data entry showed that 151 organic farms (26 organic farms converted and 125 farms in conversion) were included in the German FADN data set in 2000 and 403 organic farms (26 converted farms and 377 in conversion) were recorded in 2009. Based on the proposed indicators and with reference to guidance on best practices for organic farming (compliance with environmental principles), eight convention indicators have been identified which have been described in detail in the relevant subchapter. The authors pointed to the development of the means and the distribution of the composite indicator for the three types of farms and the development of the composite indicator of selected farms (single farms). The results suggest that the methodologies used illustrate and quantify the process of conventionalization on the basis of a large data sample, and that there are a small number of farms that are leading towards convention.

Strengths side:

Ø  Research includes extensive time period.

Ø  The methodological procedures described are enough in chapter 2(Overall approach) and chapter 3 (Indicator selection).

Weaknesses side:

Ø  Inadequate chapter Results. I suggest changing the title to "Results and discussion". The results of the work are proposed to be supplemented by a discussion (e.g. some parts from the introduction and conclusion + more).

Ø  Chapter 2 "Overall approach" and chapter 3 "Indicator selection" propose to give a common chapter "Material and methods".

 Other comments:

95-98 - I propose not to mention this section.

413 - Where are the figure 5, figure 6 and figure 7?

Conclusion - The citations should not be given in the conclusions.

Author Response

Dear Reviewer,

thank you very much for your review that helped us a lot to improve our paper. In the following, we will present our reply to your points of criticism referring to the lines of the reviewed paper under “track changes” mode.

Kind regards

The Authors

Ø  Inadequate chapter Results. I suggest changing the title to "Results and discussion". The results of the work are proposed to be supplemented by a discussion (e.g. some parts from the introduction and conclusion + more).

Now lines 548 ff: Thank you for your recommendation. We followed your suggestion and renamed the chapter. Furthermore, we extended the chapter by discussion parts, also referring to previous studies of conventionalization.

Ø  Chapter 2 "Overall approach" and chapter 3 "Indicator selection" propose to give a common chapter "Material and methods".

Now lines 111 ff: We followed again your suggestion and shifted the indicator selection to step 2 of chapter 2. We think that now the derivation of indicator selection is more comprehensible.

Other comments:

95-98 - I propose not to mention this section.

Now lines 103 ff: We agree that this section was not contributing to the presentation of our research. We reformulated it instead of completely deleting the presentation of the paper structure.

413 - Where are the figure 5, figure 6 and figure 7?

Conclusion - The citations should not be given in the conclusions.

Thank you very much for your careful reading We corrected the indication of figure numbers and deleted the citations in the conclusion.
